# Clinical Characteristics in the Acute Phase of COVID-19 That Predict Long COVID: Tachycardia, Myalgias, Severity, and Use of Antibiotics as Main Risk Factors, While Education and Blood Group B Are Protective

**DOI:** 10.3390/healthcare11020197

**Published:** 2023-01-09

**Authors:** Jose Guzman-Esquivel, Martha A. Mendoza-Hernandez, Hannah P. Guzman-Solorzano, Karla A. Sarmiento-Hernandez, Iram P. Rodriguez-Sanchez, Margarita L. Martinez-Fierro, Brenda A. Paz-Michel, Efren Murillo-Zamora, Fabian Rojas-Larios, Angel Lugo-Trampe, Jorge E. Plata-Florenzano, Marina Delgado-Machuca, Ivan Delgado-Enciso

**Affiliations:** 1School of Medicine, University of Colima, Colima 28040, Mexico; 2Clinical Epidemiology Research Unit, Mexican Institute of Social Security Institute, Villa de Álvarez 28984, Mexico; 3Molecular and Structural Physiology Laboratory, School of Biological Sciences, Universidad Autónoma de Nuevo León, San Nicolás de los Garza 66455, Mexico; 4Molecular Medicine Laboratory, Unidad de Medicina Humana y Ciencias de la Salud, Universidad Autónoma de Zacatecas, Zacatecas 98160, Mexico; 5Department of Research, Esteripharma SA de CV, Atlacomulco 50450, Mexico; 6Faculty of Human Medicine, Campus IV, Universidad Autónoma de Chiapas, Tapachula 30580, Mexico; 7Cancerology State Institute, Colima State Health Services, Colima 28085, Mexico

**Keywords:** COVID-19, long COVID, SARS-CoV-2, risk factor, tachycardia, antibiotics

## Abstract

Background: Risk factors for developing long COVID are not clearly established. The present study was designed to determine if any sign, symptom, or treatment of the acute phase, or personal characteristics of the patient, is associated with the development of long COVID. Methods: A cohort study was carried out, randomly selecting symptomatic COVID-19 patients and not vaccinated. The severity of the acute illness was assessed through the number of compatible COVID-19 symptoms, hospitalizations, and the symptom severity score using a 10-point visual analog scale. Results: After multivariate analysis, a severity score ≥8 (RR 2.0, 95%CI 1.1–3.5, *p* = 0.022), hospitalization (RR 2.1, 95%CI 1.0–4.4, *p* = 0.039), myalgia (RR 1.9, 95%CI 1.08–3.6, *p* = 0.027), tachycardia (RR 10.4, 95%CI 2.2–47.7, *p* = 0.003), and use of antibiotics (RR 2.0, 95%CI 1.1–3.5, *p* = 0.022), was positively associated with the risk of having long COVID. Higher levels of education (RR 0.6, 95%CI 0.4–0.9, *p* = 0.029) and type positive B blood group (B + AB, RR 0.44, 95%CI 0.2–0.9, *p* = 0.044) were protective factors. The most important population attributable fractions (PAFs) for long COVID were myalgia (37%), severity score ≥8 (31%), and use of antibiotics (27%). Conclusions: Further studies in diverse populations over time are needed to expand the knowledge that could lead us to prevent and/or treat long COVID.

## 1. Introduction

Coronavirus disease (COVID-19) is caused by the severe acute respiratory syndrome coronavirus 2 (SARS-CoV-2) [1,2]. Although COVID-19 affects the respiratory system mainly, it is a truly multisystem disease [3]. Extrarespiratory complications are uncommon and include dysregulated immune responses, metabolic dysfunction, and adverse effects on multiple organ function, notably increased cardiorenal inflammation, although they can affect the nervous, endocrine, and musculoskeletal systems as well [4,5,6,7]. Previous studies reported that the most common initial symptoms are headache, fatigue, myalgia, sore throat, and cough [3]. Other very common signs or symptoms include arthralgia, fever, chills, rhinorrhea, nausea, conjunctivitis, anosmia, ageusia, and dizziness, among others [3]. Most infections are self-limiting, but 20% of symptomatic, infected, unvaccinated adults need hospitalization [3]. Vaccinated individuals are 10 times less likely of suffering severe/critical COVID-19 [5]. However, the proportion of patients who develop severe COVID-19 and require hospitalization may vary by region and country, according to the characteristics of the population, diagnostic strategies, or therapeutic or vaccination schemes used [6,8].

COVID-19 is a short-term illness, and in general, the patients return to their usual state of health 12–14 days after receiving a positive test result [9]. However, it has been reported that some patients still have symptoms for weeks or even months after infection. The term “Chronic COVID” or “Long COVID” describes symptoms persisting for more than 12 weeks after infection. An analysis of 41 reports estimates a long COVID global prevalence of approximately 43% [10], although there are studies that have reported a prevalence of up to 80% [11]. In Mexico, the reported prevalence is 76% [12].

The most frequent symptoms of the long COVID are walking/exercise intolerance, fatigue, and myalgias, among others [13,14]. Risk factors for developing long COVID are not clearly established, although some studies have associated it with female sex, respiratory problems at the onset of illness [13], longer duration or critical forms of the acute disease [4], smoking, comorbidities, lower degree of education, and other socioeconomic deprivations [15]. Nevertheless, the presence of signs or symptoms that by themselves can be directly associated with the development of long COVID, regardless of the disease severity, has not been analyzed. Likewise, the use of antibiotics [4], and blood type are suspected to be associated with the evolution of the disease [16], although their association with the development of chronicity has not been demonstrated.

Therefore, the present study was designed to determine if any sign, symptom, or treatment of the acute phase is associated with long COVID in a Mexican population. Universal variables, comorbidities, and blood type were also analyzed.

## 2. Materials and Methods

### 2.1. Study Design

A cohort analysis was conducted. Patients who went through the acute phase of COVID-19 between June and October 2020 were included. The patients were evaluated for symptom persistence after 3 and 6 months (4.2 + 1.2 months). In the present study, we examined the clinical and social characteristics, as well as present signs and symptoms in the acute phase of COVID-19, in association with experiencing long COVID. All the data were obtained through a medical evaluation as part of their health care. The patients were seen for a follow-up medical evaluation between 3 and 6 months after suffering from COVID-19. The study was approved by the ethics committee of the General Hospital of Zone 1, IMSS Colima. Moreover, informed consent was obtained from all subjects involved in the study.

### 2.2. Study Subjects

The inclusion criteria were the following: men and nonpregnant women >18 years of age, presenting with COVID-19 confirmed with a positive diagnosis of SARS-CoV-2 by RT-PCR, medical consultation in the General Hospital (Number 1 of the IMSS, Colima, Mexico) due to their acute illness, and that they were indicated for at-home ambulatory or hospital treatment. The patients received the usual medical attention and continued with the usual treatment prescribed by their family doctor or specialist. Asymptomatic patients and pauci-symptomatic patients were excluded, as previously defined [3,17].

### 2.3. Measures and Follow-Up

For patients with acute illness, the data was collected at the emergency room by the admitting physician and entered in medical records. Universal variables (age, sex, education, etc.), pathological/non-pathological personal history (blood type, skin type, and comorbidities) and signs and symptoms related to COVID-19 were collected (fever, tachycardia, chills, cough, respiratory distress, muscle pain, arthralgia, sore throat, conjunctivitis, anosmia, dysgeusia, dizziness, headache, rhinorrhea, diarrhea, chest or abdominal pain, nausea, vomiting, anorexia, diaphoresis, paresthesia, asthenia, and adynamia). Tachycardia is defined as a heart rate greater than 100 beats per minute with peripheral oxygen saturation (SpO2) >90% and without fever. Other collected data included: treatment and admission to a hospital ward, or other clinical data in subsequent medical evaluations. The Fitzpatrick skin scale test was used [18]. The patient overall self-assessment or symptom severity score was collected for each patient routinely, as previously described [3,19]. For this, the following question was asked: “Considering all the ways in which illness and health conditions may affect you at this time, please indicate below how are you feeling?” Self-assessment using a 0–10-point VAS (from ‘very well’—score of 0—to ‘very poorly’—score of 10), and has been used in other COVID-19 clinical trials as well [3,19]. The diagnosis dates of the patients were between June and October 2020. The discharged patients were invited to participate in the study through a telephone call, between three to six months after their illness. If they accepted, they were given an appointment for a medical evaluation. Patients were randomly selected from the institutional database. If someone did not answer on a second attempt, the previous patient on the list was selected. Data from the medical records collected in the acute phase of COVID-19 was used, while the symptoms of long COVID were evaluated at the new medical appointment to which the included subjects were invited. The symptoms of long COVID were very similar to those previously reported by other studies, and subjects were included in the case group with the presence of just one symptom [13,14,15].

### 2.4. Sample Size

The sample size calculation was based on the number of patients that had a severity symptom score ≥8 in the acute phase and that developed chronic symptoms. The sample size was calculated by estimating that 76% of patients with a severity score ≥8 would develop long COVID, expecting that those with a severity score of 7 or less would develop it in half this proportion (33%). Those figures were based on local data from Mexico, in which 76% of confirmed patients developed long COVID [11]. Twenty patients from each group were needed to reach the required power (0.8). Once the study was completed, the statistical power of the presence of tachycardia or severity score ≥8, as predictors of long COVID, was calculated, resulting in 98.9% and 99.9%, respectively.

### 2.5. Statistical Analysis

The data were presented as the mean ± standard deviation. The normal distribution of the data was verified with the Kolmogorov–Smirnov test. The categorical values were compared between groups, using Fisher’s exact test or the likelihood ratio chi-square test. The numerical data were compared, utilizing the Student’s *t*-test. Univariate binary logistic regression analyses were employed to determine the probability of developing long COVID (binomial outcome: yes or no) with the presence of each symptom or sign observed in the acute phase, or with the presence of different general or clinical characteristics of the patients. The data were summarized as relative risks (RRs) with 95% confidence intervals (CIs) and *p*-values. Significant predictors were subsequently added to the multivariable model, and with a forward stepwise logistic regression the most parsimonious model was identified. The probability used for the stepwise regression was set at 0.25 for entry of variables and 0.20 for removal. Binomial regression is considered the most adequate choice for estimating RRs in multivariate analyses [20,21,22]. The adjusted population attributable fractions (PAF) were calculated for the main predictors for long COVID, using the Miettinen formula and considering the adjusted RR, according to previous reports [23]. PAF was expressed as a percentage and Pearson correlations were used for bivariate analyses (severity symptom score, and number of symptoms in acute or long disease). The area under the receiver operating characteristic (ROC) curve, confidence interval, cut-off point, sensitivity, and specificity of the symptom severity score in the acute phase were calculated to discriminate subjects who would develop long COVID.

Data were analyzed by using SPSS Statistics version 20 software (IBM Corp., Armonk, NY, USA), except for the calculation of statistical power and sample size, which were performed with the online software CinCalc version 1 (https://clincalc.com/stats/Power.aspx; (accessed on 25 April 2022)) [24]. A *p* < 0.05 was considered statistically significant. 

## 3. Results

### 3.1. Patient Characteristics

Three hundred and eleven patients were included and analyzed. A total of 115 (37%) did not present chronic symptoms (control group) and 196 (63%) patients had at least one persisting symptom which allowed for them to be classified as long COVID. Most of the patients with long COVID had only one chronic symptom (69.5%) and only a very small proportion had three or more symptoms (4.6%). The most frequent symptom was exercise intolerance (15.7%), followed by muscle pain (13.7%), joint pain (13.7%), fatigue (12.7%), headache (11.2%), and shortness of breath (10.2%) (Table 1).

The recovery time from the acute illness was 4.2 + 1.2 vs. 4.0 + 1.3 months for patients in the control group and those with long COVID, respectively (*p* = 0.183) (Table 2). In acute COVID-19, the symptom severity score; according to a self-assessment 10-point visual analog scale, was 6.8 + 2.6, and 27.5% required hospitalization. The reported median number of COVID-19-compatible symptoms was 11.8 + 4.8. None of the patients was asymptomatic or pauci-symptomatic. The main clinical characteristics, prescribed drugs, and overview of the symptoms in acute disease are shown in Table 2 and Table 3.

### 3.2. Risk Factors for Entering the Chronic Phase

Differences in some variables were observed between patients who had long COVID vs those who did not. Patients with long COVID had a higher degree of severity in the acute phase of the disease (higher symptom severity score, higher number of symptoms, and a greater proportion of hospitalized patients) (see Table 2 and Table 3). They also received a greater number of treatments, although antibiotics were the only group of drugs that were significantly more frequent in patients who would have long COVID (Table 1). There were also differences between blood type groups (Table 1). Patients with long COVID also had a greater proportion of signs or symptoms in the acute phase, which included tachycardia, myalgia, sore throat, nausea, conjunctivitis, dizziness, diarrhea, chest pain, dyspnea, abdominal pain, and diaphoresis (see Table 3). The number of symptoms in long COVID was positively correlated with the number of symptoms (r = 0.28, *p* < 0.001) and the symptom severity score (r = 0.27, *p* < 0.001) of the acute phase.

The area under the ROC curve for symptom severity score as a predictor of long COVID was 0.70 (95% CI 0.63–0.75, *p* <0.001) (Figure 1). A symptom severity score of ≥8 showed a sensitivity of = 0.64 and a specificity of = 0.87, which suggests that this marker is an adequate “rule-in” test for long COVID [25].

In the univariate models, several factors in the acute illness were positively associated with the risk of having long COVID, such as: chest pain (RR 1.8, 95% CI 1.1–3.0, *p* = 0.008), dyspnea (RR 2.2, 95% CI 1.4–3.5, *p* = 0.001), sore throat (RR 1.6, 95% CI 1.0–2.6, *p* = 0.024), conjunctivitis (RR 1.9, 95% CI 1.0–3.6, *p* = 0.047), tachycardia (RR 10.1, 95% CI 2.3–43.3, *p* = 0.002), nausea (RR 2.9, 95% CI 1.2–7.0, *p* = 0.012), severity score ≥8 (RR 3.32, 95% CI 2.0–5.4, *p* < 0.001), hospitalization (RR 2.6, 95% CI 1.4–4.6, *p* = 0.001), and use of antibiotics (RR 1.6, 95% CI 1.0–2.6, *p* = 0.033) (Table 4). These associations remained in the multivariate model for severity score ≥8 (RR 2.0, 95% CI 1.1–3.5, *p* = 0.022), hospitalization (RR 2.1, 95% CI 1.0–4.4, *p* = 0.039), myalgia (RR 1.9, 95% CI 1.08–3.6, *p* = 0.027), tachycardia (RR 10.4, 95% CI 2.2–47.7, *p* = 0.003), and use of antibiotics (RR 2.0, 95% CI 1.1–3.5, *p* = 0.022). A high level of education was a protective factor in the multivariate model (RR 0.6, 95% CI 0.4–0.9, *p* = 0.029). All other factors included in the multivariate model were not significantly associated with long COVID (see Table 4). In addition, the population attributable fraction (PAF) was calculated, which was defined as the proportional reduction in the occurrence of long COVID in the study population, if exposure to a risk factor was eliminated in an alternative ideal exposure scenario [23]. The higher PAF for long COVID were myalgia (37%), severity score ≥8 (31%), and use of antibiotics (27%); hospitalization (18%) and the presence of tachycardia (14%) were less relevant at a population level.

The proportion of ABO blood type was different in patients with long COVID vs without (Table 2). Type O was very similar in both groups (59.5% vs. 59.0%), but positive type B patients (B and AB groups) were less frequent in long COVID (15.3% vs. 8.0%, *p* < 0.001, without vs with long COVID, respectively). In a multivariate model adjusting with the main factors associated with long COVID (severity score ≥8, hospitalized patients, and tachycardia), positive type B patients presented nearly two times less risk of suffering long COVID, in comparison with patients with O/A blood type (RR 0.44, 95% CI 0.20–0.97, *p* = 0.044).

## 4. Discussion

Severe COVID-19 was confirmed as a predicting factor for long COVID. However, some signs or symptoms of the acute phase are also independent risk factors, such as tachycardia, with a 10 times higher probability for long COVID, for patients who present said symptom.

Previously, sex (female), respiratory symptoms at the onset, and the severity of the illness were associated with long COVID [13]. Although the female sex was more frequent in the present study in patients with long COVID, there was no significant difference (59.9 vs. 53.0%, *p* = 0.144), just as being a woman was not a predicting factor in the multivariate analysis. Higher COVID-19 severity associated with long COVID has been previously expressed in a longer disease duration, as well as in the presentation of critical forms of the disease (higher symptom numbers, hospitalizations, and intensive care unit admissions) [4,13,14,15]. This agrees with the results found in the present study, where a higher number of symptoms in the acute phase, more hospitalizations, and a higher severity symptom score were associated with long COVID. Furthermore, it was found that the higher the severity symptom score, the higher the number of chronic symptoms. The severity symptom score is a self-assessment which uses a 10-point VAS. It is easy to obtain through a single question and it can help guide a patient’s prognosis. It is the first time this scale has been used to establish the association between severity in the acute phase and the development of long COVID, although it has been used in clinical trials of COVID-19 [3]. A score of 8 or higher in the severity symptom score helps to predict the apparition of chronic symptoms with an 87% specificity, although its low sensibility (64%) shows that it is not very useful to rule out patients who will have chronic disease (high probability of obtaining false negatives). This supports the hypothesis that other factors are also very important in the genesis of long COVID. 

Tachycardia was, by far, the factor which generates a higher probability for the genesis of chronic symptoms of COVID-19. Inappropriate sinus tachycardia can be triggered by fever and hypoxia, but in COVID-19 infection [26], it could also be related to systemic inflammation (with high levels of protein C reactive), pathological lung changes, and extended hospital stays [27]. Tachycardia has been suggested as an acute phase disease severity prognostic factor, as survivors presented less tachycardia than non-survivors [28]. Hsieh JYC et al. (2021) have previously postulated that “although sinus tachycardia reflects a longer and possibly more inflammatory clinical course among patients with mild COVID-19 infection, it can happen in the absence of significant pulmonary, thyroid, and hematological dysfunction” (without thromboembolism) [27]. Due to the aforementioned, tachycardia could also occur independently of the perceived disease severity [27]. It has been reported that tachycardia in mild COVID-19 infection may result from reversible physiological changes, such as dehydration or mild inflammatory response. However, the findings about its important association with long COVID could reflect pathophysiological alterations of long duration and are still unknown to us. By itself, an inappropriate sinus tachycardia is a common observation in patients with post-COVID-19 [29], which has been reported as possibly due to an injury in the autonomic nervous system [29]. Due to this, it is likely that tachycardia in the chronic and acute phases of COVID has a different pathophysiology and it remains unknown whether they are related or not.

Several symptoms of COVID-19 had a higher proportion in patients who would present chronic symptoms. However, in the present research, myalgia was the only symptom significantly associated with long COVID, even after adjusting with multivariate analyses. Myalgia is a common phenomenon in viral infections and may be due to generalized inflammation with a cytokine response. However, the myalgia and fatigue experienced by COVID-19 patients may be of longer duration than in other viral infections, and they generally respond poorly to conventional analgesics. It has been postulated that the musculoskeletal pain caused by COVID-19 may be generated by completely different mechanisms than other viral infections [30]. Regardless of muscle inflammation, COVID-19 can increase lactate levels, in addition to lowering muscle oxygenation and pH, factors that can contribute to myalgia [30]. Due to these mechanisms, myalgia in COVID-19 may have an ischemic component, with a mechanism similar to sickle cell disease [30]. In addition, during hypoxic ischemia, an increase in cytokines and growth factors, coupled with microvascular changes within the muscles, can cause overexpression of different receptor molecules (purinergic receptors such as P2RX4, P2RY1, P2RX3, P2RX5, or others as transient receptor potential cation channel subfamily V member 1, and acid-sensing ion channel subunit 1) in the dorsal root ganglia, which in turn modulate sympathetic reflexes and pain [30,31]. These molecular changes, which may take a long time to go back to normal levels, are compatible with the genesis of myalgia and chronic fatigue due to COVID-19. Fatigue and myalgia in long COVID are clinically similar in certain aspects to the ischemic pain developed in other pathologies as well, such as juvenile fibromyalgia, complex regional pain syndrome, or peripheral vascular disease [31].

Regarding the use of antibiotics and its association with long COVID, there is only one previous report that found said association [4]. It could be hypothesized that this association is because patients with a more severe condition are the ones who receive antibiotics more frequently. However, the association between antibiotics and long COVID was found even after making statistical adjustments, which shows that it could be an independent factor in the severity of the disease. Ahmed Samir Abdelhafiz et al. (2022) report that the use of antibiotics generates twice the probability of developing long COVID, which is in line with the results found in the present study. Ahmed Samir et al. suggest that the side effects of these drugs may contribute to delayed recovery in those patients [4]. However, we hypothesize that the use of antibiotics generates an altered gut microbiome composition, which has been strongly associated with persistent symptoms in patients with COVID-19 [32]. For this reason, likewise Ahmed Samir et al., the results of the present investigation also reinforce the recommendation for limiting the use of these drugs to select cases according to international guidelines [32]. Furthermore, the use of probiotics is recommended to reestablish the intestinal bacterial microbiota, during and after the acute phase of COVID-19. The administration of probiotics has been postulated as a complementary treatment to fight COVID-19, because they, and their metabolites, have known antiviral properties [33]. Further research is needed to determine if probiotics play a preventive and/or therapeutic role in long COVID.

Regardless of the risk that the aforementioned variables represent for the genesis of long COVID to those who suffer from them, it is important to measure their relevance at a population level. If exposure to risk factors were removed, in an alternate ideal scenario (PAF), the proportional decrease of long COVID would show 37% for myalgia, 31% for severity score ≥8, and 27% for antibiotic use. Hospitalization (PAF = 18%) and tachycardia (PAF = 14%) had a lower relevance at a population level due to their lower prevalence among cases, despite being the most predictive factors (higher RR) for long COVID. The PAF assumes that there is a perfect intervention which eradicates the exposure, which is often unrealistic [23]. However, therapeutic strategies could be done to help reduce the incidence and/or intensity of these factors during COVID-19 (exposure partially removed), providing a potentially beneficial impact at a population level in the prevalence of long COVID. This is confirmed by the studies that mention how vaccination campaigns in diverse populations reduce hospitalizations, and at the same time have helped reduce the prevalence of long COVID, although there are studies that show that its effect on the frequency of this pathology does not exist, or it is only a slight effect [34,35]. This issue remains controversial [15,35].

Many studies report that blood type might be a factor in the presentation of the infection with SARS-CoV-2 [36]. The results are controversial, with the suggestion that type O could be a protective factor against acute infection [37], but a risk factor for long COVID [36]. The differences could be due to variations between the acute and chronic pathologies, or to variations peculiar to the analyzed population. The present study found that B-positive blood type (B + AB groups) reduces 2.3 times the risk of developing long COVID in comparison to non-B groups (O + A groups). This is very similar to what Díaz-Salazar S. et. Al. (2022) reported that O type is a factor that increases the risk of long COVID [37]. However, there are two differences with the Díaz-Salazar report: (1) Type O is the more abundant blood type in the Mexican population, so it was used as the reference group, which results in B-positive blood groups as protective factors (in comparison to non-B groups, such as O, which would have a higher risk); and (2) in the present study, groups A and B had very different values in the proportion of individuals who developed long COVID. Group A had a higher prevalence than group O in the chronic disease, so it did not follow to put groups A and B together. There are previous studies that have found that A and B types can have different effects on diverse pathological factors of COVID-19 [38]. There are still few studies about the association between blood types and long COVID, and the results are controversial [39], so more studies are needed to obtain more solid conclusions.

We also found that a higher level of education creates a protection against long COVID. This is in line with what the study of Ahmed Samir Abdelhafiz (2022) reported, that a lower education level was significantly associated with the presence of post-COVID symptoms [4]. This association could reflect a less efficient medical treatment for people who live in less favorable social conditions. However, this is a hypothesis that would need to be analyzed in further research.

Early treatment of the acute phase to reduce signs and symptoms through anti-inflammatories, antivirals, or having a complete vaccination schedule, among other strategies, could contribute to reducing the incidence of long-term COVID. Any measure that reduces inflammation or viremia [40], such as not administering antibiotics when they are unnecessary, or re-establishing the intestinal microbiota after they are administered [32,33], may reduce sequelae incidence. Additionally, mild symptoms do not exclude the possibility of sequelae, so a clinician-guided treatment is necessary.

The present analysis has several limitations. The study was made in non-vaccinated people infected with COVID-19 during 2020, so the results of future research could vary due to vaccination and reinfection status, or to the different virus variants, as well as individual patient variations in antibody development after acute infection or vaccination [41,42]. Another limitation of the study is that mental health symptoms, such as depression, anxiety, etc., were not evaluated and have also been associated with sequelae of COVID [43]. Long COVID requires further research among diverse populations over time since variations in the incidence of the syndrome and its associated factors might be linked to region-specific treatment options and vaccination schedules. It is also important to mention that the risk factors mentioned here denote association, but they are not necessarily causal factors, so future studies are necessary.

## 5. Conclusions

The severity of COVID-19 (higher number of symptoms, more hospitalizations, and a higher severity symptom score in the acute phase) and the presence of myalgia, tachycardia, or antibiotic use are the main risk factors for long COVID, while higher education and blood type B are protectors. Further studies in diverse populations over time are needed to increase the knowledge that will allow us to prevent and/or treat long COVID.

## Figures and Tables

**Figure 1 healthcare-11-00197-f001:**
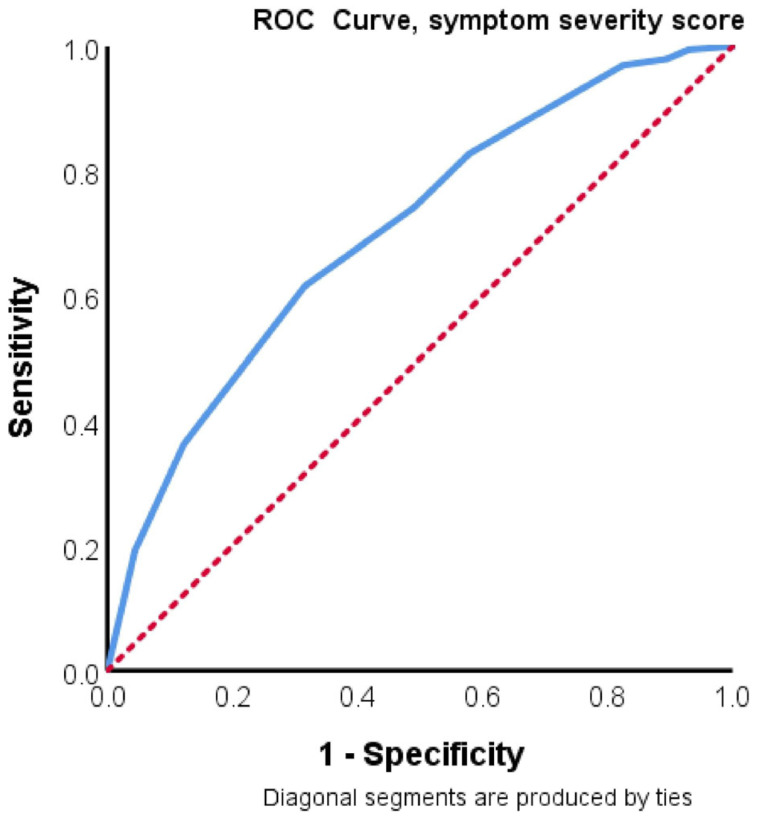
ROC curve of symptom severity score as a predictor of long COVID.

**Table 1 healthcare-11-00197-t001:** Main symptoms/signs/complaints of long COVID of the participating subjects.

	Proportion with Respect to Patients with	
	Long Covid n = 197	All n = 311
No. symptoms		
0	0.0%	36.7%
1 or more	100%	63.3%
1	69.5%	44.1%
2	25.9%	16.4%
3 or more	4.6%	2.9%
Exercise intolerance	15.7%	10.0%
Muscle pain	13.7%	8.7%
Joint pain	13.7%	8.7%
Fatigue	12.7%	8.0%
Headache	11.2%	7.1%
Shortness of breath	10.2%	6.4%
Chest pain	7.6%	4.8%
Walking intolerance	6.6%	4.2%
Dizziness	5.1%	3.2%
Weakness	4.6%	2.9%
Loss of smell	4.6%	2.9%
Sore throat	4.1%	2.6%
Cough	3.6%	2.3%
Hair loss	3.6%	2.3%
Paresthesia	3.6%	2.3%
Loss of taste	3.0%	1.9%
Diarrhea	3.0%	1.9%
Palpitation	2.0%	1.3%
Sleep difficulty	1.5%	1.0%
Brain fog	1.5%	1.0%
Abdominal pain	1.5%	1.0%
Excess sputum	1.0%	0.6%
Excess sweating	1.0%	0.6%

**Table 2 healthcare-11-00197-t002:** Main clinical characteristics of the participating subjects at the time of enrollment and prescribed drugs.

Clinical Characteristic	All	Long COVID		*p*
		No	Yes	
Female (%)	57.4%	53.0%	59.9%	0.144 *
Age (years)	76.4%	44.5 ± 13.3	46.3 ± 13.1	0.236 **
BMI	70.2%	28.4 ± 5.6	29.3 ± 5.4	0.139 **
Skin type test ^^^				0.940 ***
Type I	1.0%	0.9%	1.0%	
Type II	11.1%	11.5%	10.8%	
Type III	38.8%	38.1%	39.2%	
Type IV	39.4%	38.9%	39.7%	
Type V	9.4%	10.6%	8.8%	
Type VI	0.3%	0.0%	0.5%	
Education				0.333 ***
None	2.9%	1.7%	3.6%	
Elementary	25.3%	20.9%	27.9%	
High school	26.1%	26.5%	25.8%	
Bachelor’s degree or more	46.2%	51.3%	43.1%	
Diabetes	19.9%	24.3%	17.3%	0.087 *
High blood pressure	23.1%	18.3%	25.9%	0.079 *
Asthma	7.1%	5.2%	8.1%	0.233 *
Smoking	42.9%	47.0%	40.6%	0.165 *
Alcoholism	19.9%	20.4%	19.7%	0.500 *
Cancer	4.5%	5.2%	4.1%	0.419 *
Blood type				0.031 ***
O	59.2%	59.5%	59.0%	
A	30.1%	25.2%	33.0%	
B	9.7%	12.6%	8.0%	
AB	1.0%	2.7%	0.0%	
RH+	96.9%	97.2%	96.8%	0.557 *
Time evolution	4.1 ± 1.3	4.2 ± 1.2	4.0 ± 1.3	0.183 **
Symptom severity	6.8 ± 2.6	5.7 ± 2.7	7.4 ± 2.1	<0.001 **
No. of treatments	2.6 ± 1.5	2.3 ± 1.4	2.7 ± 1.5	0.019 **
Para/NSAIDs	92.0%	90.4%	92.9%	0.290 *
Ivermectin	11.3%	11.4%	11.3%	0.556 *
Antibiotics	49.7%	41.7%	54.3%	0.021 *
Antivirals	21.2%	19.1%	22.4%	0.294 *
Steroids	19.0%	14.0%	21.8%	0.060 *
Anticoagulants	24.5%	19.3%	27.6%	0.067 *
Vitamins	36.8%	32.5%	39.3%	0.140 *
Chlorine dioxide	2.3%	1.8%	2.6%	0.491 *
Zinc	1.9%	0.9%	2.6%	0.282 *

Percentages or averages and standard deviations are shown. * Fisher’s exact test analysis. ** Student’s *t*-test analysis. *** Likelihood ratio chi-square test. BMI: Body mass index. ^^^ Skin types: using the Fitzpatrick skin type test, including type I (palest; freckles), type II (light colored but darker than fair), type III (golden honey or olive), type IV (moderate brown), type V (dark brown), and type VI (deeply pigmented dark brown). Time evolution: months since acute phase. Para/NSAIDs: paracetamol/nonsteroidal anti-inflammatory drugs; symptom severity: patient overall self-assessment, using a visual analog scale from 0 to 10 points.

**Table 3 healthcare-11-00197-t003:** Proportion of patients who presented diverse signs and symptoms of the disease in the acute phase according to whether they developed long COVID or not.

Symptoms/Signs (Acute Phase)	All n = 311	Presented Long COVID		*p* *
		No n = 115	Yes n = 196	
Headache	76.3%	73.9%	77.7%	0.268
Myalgia	69.9%	61.7%	74.6%	0.017
Sore throat	50.0%	41.7%	54.8%	0.012
Cough	61.4%	55.7%	64.8%	0.070
Arthralgia	66.3%	60.9%	69.5%	0.075
Fever	75.8%	72.2%	77.9%	0.156
Chills	47.3%	43.5%	49.5%	0.182
Rhinorrhea	36.9%	31.3%	40.2%	0.074
Nausea	12.5%	6.1%	16.2%	0.006
Conjunctivitis	18.8%	13.0%	22.3%	0.030
Anosmia	47.4%	42.6%	50.3%	0.118
Dysgeusia	44.9%	39.1%	48.2%	0.075
Dizziness	17.3%	12.2%	20.3%	0.045
Vomiting	20.8%	19.1%	21.8%	0.339
Diarrhea	44.2%	37.4%	48.2%	0.041
Chest pain	55.9%	46.1%	61.7%	0.005
Dyspnea	59.3%	47.0%	66.5%	0.001
Abdominal pain	33.3%	27.0%	37.1%	0.044
Tachycardia	10.3%	1.7%	15.2%	<0.001
Diaphoresis	11.5%	7.0%	14.2%	0.037
Paresthesia	4.2%	1.7%	5.6%	0.085
Asthenia	5.8%	3.5%	7.1%	0.140
Adynamia	4.2%	2.6%	5.1%	0.228
Hospitalization	27.5%	17.0%	33.7%	0.001
No. symptoms ^	10.8 ± 4.7	9.2 ± 4.2	11.8 + 4.8	<0.001
Severity score	6.8 ± 2.5	5.7 ± 2.8	7.5 + 2.1	<0.001
Severity score ≥8	50.5%	32.2%	61.2%	<0.001

* Comparison between those who presented long COVID vs those who did not. ^ Number of COVID-19-compatible symptoms. All analyses were performed with Fisher’s exact test, except for the variables (1) number of symptoms, and (2) severity score, showing media and standard deviation, which were analyzed using Student’s *t*-test analysis.

**Table 4 healthcare-11-00197-t004:** Univariate and multivariate logistic regression analyses of factors associated with long COVID in the acute phase of the disease.

	Univariate Analysis								
Variable *	cRR	95% C.I.		*p* Value	Variable *	cRR	95% C.I.		*p* Value
		Lower	Upper				Lower	Upper	
Female (%)	1.32	0.83	2.10	0.238	Myalgia	1.82	1.11	2.98	0.017
Age (years)	1.01	0.99	1.03	0.139	Arthralgia	1.46	0.90	2.37	0.119
BMI	1.02	0.98	1.07	0.244	Rhinorrhea	1.47	0.90	2.40	0.118
Skin typing test ^^^	0.98	0.75	1.29	0.920	Sore throat	1.69	1.06	2.69	0.024
Blood type ^π^	0.82	0.59	1.14	0.259	Conjunctivitis	1.91	1.00	3.62	0.047
RH+	0.71	0.18	2.83	0.637	Tachycardia	10.15	2.37	43.31	0.002
Education ^†^	0.78	0.60	1.02	0.075	Diaphoresis	2.21	0.97	5.04	0.058
Diabetes	0.64	0.36	1.13	0.132	Anosmia	1.36	0.85	2.16	0.193
HBP	1.56	0.88	2.76	0.125	Dysgeusia	1.44	0.90	2.31	0.120
Smoking	0.77	0.48	1.22	0.275	Nausea	2.99	1.27	7.02	0.012
Alcoholism	0.97	0.60	1.56	0.905	Dizziness	1.83	0.95	3.54	0.070
Cancer	0.77	0.26	2.28	0.642	Paresthesia	3.34	0.72	15.34	0.121
Asthma	1.60	0.61	4.22	0.337	Asthenia	2.12	0.68	6.61	0.194
Severity score ≥8	3.32	2.04	5.40	<0.001	Adynamia	1.99	0.53	7.40	0.301
Fever	1.36	0.80	2.3 1	0.252	Hospitalization	2.60	1.46	4.62	0.001
Cough	1.46	0.91	2.34	0.111	Para/NSAIDs	1.37	0.60	3.14	0.450
Chest pain	1.88	1.18	3.00	0.008	Antivirals	1.22	0.69	2.17	0.490
Dyspnea	2.24	1.40	3.59	0.001	Antibiotics	1.65	1.04	2.64	0.033
Headache	1.22	0.71	2.09	0.453	Ivermectin	0.98	0.47	2.04	0.974
Diarrhea	1.56	0.97	2.49	0.064	Steroids	1.71	0.91	3.20	0.094
Vomiting	1.18	0.66	2.09	0.572	Anticoagulants	1.59	0.90	2.78	0.105
Abdominal pain	1.59	0.96	2.63	0.069	Vitamins	1.34	0.82	2.18	0.230
	Multivariate analysis								
Variable *	aRR	95% C.I.		*p* Value	Variable *	aRR	95% C.I.		cRR
		Lower	Upper				Lower	Upper	
Education ^†^	0.69	0.49	0.96	0.029	Myalgia	1.98	1.08	3.63	0.027
Diabetes	0.64	0.32	1.28	0.207	Rhinorrhea	1.48	0.83	2.67	0.186
Smoking	0.66	0.38	1.15	0.143	Tachycardia	10.43	2.28	47.79	0.003
Severity score ≥8	2.00	1.11	3.53	0.022	Hospitalization	2.16	1.04	4.47	0.039
Diarrhea	1.58	0.89	2.83	0.119	Antibiotics	2.00	1.11	3.53	0.022
Vomiting	0.52	0.25	1.08	0.079					

* Clinical characteristics or treatments in acute COVID-19. cRR: crude relative risk, aRR: adjusted relative risk. HBP: High blood pressure. *p* < 0.05 means statistically significant. CI: confidence interval. Important predictors in the univariate model were subsequently added to the multivariate model and a forward stepwise logistic regression identified the most parsimonious model. The probability used for the stepwise regression was set at 0.25 for entry of variables and 0.20 for removal. ^^^ Skin types: Using the Fitzpatrick skin type test, listed as: 1 = type I, 2 = type II, 3 = type III, 4 = type IV, 5 = type V, and 6 = type VI. ^π^ Blood type: listed as: O = 0, A = 1, B = 2, and AB = 3. ^†^ Education: listed as: none = 0, elementary = 1, high school = 2, and 3 = bachelor’s degree or more. Para/NSAIDs: paracetamol/nonsteroidal anti-inflammatory drugs; symptom severity: patient overall self-assessment, using a visual analog scale from 0 to 10 points.

## Data Availability

All relevant data appears in the paper. The datasets used and/or analyzed in the current study are available from the corresponding author upon reasonable request.

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
