# Peer review of "Clinical Characteristics in the Acute Phase of COVID-19 That Predict Long COVID: Tachycardia, Myalgias, Severity, and Use of Antibiotics as Main Risk Factors, While Education and Blood Group B Are Protective"

_healthcare, 2023, doi:10.3390/healthcare11020197_

Round 1

Reviewer 1 Report

Very well conducted study, with relevant clinical implications. May I suggest to use the term "predictive" instead of "risk" factor to describe the association of certain clinical parameters with the future incidence of long Covid . For example, tachycardia is predictive (not necessarily a risk factor) for long Covid. On the other hand, when presenting time results, please state that this is expressed in "months" units.

Strenghts of the article are related to the good clinical evaluation and follow-up, allowing relevant predictive information.

Limitations may be related to the region-specific treatment availability. Discussion may comment the fact that a higher use of antivirals and anticoagulants, among other factors, might have changed the incidence of long Covid.

Author Response

The paper “Clinical characteristics in the acute phase of COVID-19 that predict long COVID: tachycardia, myalgias, severity, and use of antibiotics as main risk factors, while education and blood group B are protective”

Manuscript ID: healthcare-2058257

All the changes are listed below, following the reviewers’ suggestions. All the suggestions were taken into consideration and the necessary changes were made in the text.

Reviewer 1

Comment: Very well conducted study, with relevant clinical implications. May I suggest to use the term "predictive" instead of "risk" factor to describe the association of certain clinical parameters with the future incidence of long COVID. For example, tachycardia is predictive (not necessarily a risk factor) for long Covid.

Answer: Thank you for your observation. In the sections where the role of tachycardia, or the use of antibiotics, for the development of long COVID was analyzed, the word "risk" was replaced by "probability." This does not alter the results' interpretation and follows what the reviewer suggests.

However, in the title or in the sections where the role of several factors is described simultaneously, the word "risk factor" was kept, considering that epidemiological terminology is being used, where "Risk factors are characteristics at the biological, psychological, family, community, or cultural level that precedes and is associated with a higher probability of negative outcomes." Epidemiologists often use the term "risk factor" to indicate a factor associated with a given outcome. However, a risk factor is not necessarily a cause. The term risk factor includes proxies for underlying causes. In the discussion section, it is clarified that risk factors denote association, but may not be causal factors, so further studies are needed.

Comment: On the other hand, when presenting time results, please state that this is expressed in "months" units.

Answer: Thanks for the observation. The time unit "months" was added in the results section when the evolution time of the patients was mentioned.

Comment:Strenghts of the article are related to the good clinical evaluation and follow-up, allowing relevant predictive information.

Answer: Thank you very much for your comments

Comment: Limitations may be related to the region-specific treatment availability. Discussion may comment the fact that a higher use of antivirals and anticoagulants, among other factors, might have changed the incidence of long Covid.

Answer: Thank you for your comment. Your idea has been added to the discussion section. "Long COVID requires further research among diverse populations over time since variations in the incidence of the syndrome and its associated factors might be linked to region-specific treatment options and vaccination schedules."

Reviewer 2 Report

PLEASE ELABORATE SOME INTERNAL ORGAN DAMGE AND MORE INFORMATION  IN INTRODUCTION LINE 4.

THese ARTICLE CAN BE HELPFUL:

-Kidney diseases and COVID-19 infection: causes and effect, supportive therapeutics and nutritional perspectives. Hassan Askari, Nima Sanadgol, Asaad Azarnezhad, Amir Tajbakhsh, Hossein Rafiei, Ali Reza Safarpour, Seyed Mohammad Gheibihayat, Ehsan Raeis-Abdollahi, Amir Savardashtaki, Ali Ghanbariasad, Navid Omidifar.Heliyon,Volume 7, Issue 1,2021,e06008,ISSN 2405-8440, https://doi.org/10.1016/j.heliyon.2021.e06008.

If we could have serologic data of patients to have a view of possible immunologic response on sign and symptoms, the article would have been stronger.

Such data for comparison can be found in these serial articles:

-Prevalence of anti-sars-cov-2 antibody in hospital staff in double-center setting: A preliminary report of a cohort study from Iran. Bagheri Lankarani K , Honarvar B, Omidifar N, Pakdin M, Moghadami M, et al. Shiraz E-Med J.22(3):e112681. doi: 10.5812/semj.112681.

-SARS-CoV-2 Antibodies in the Public Hospital Staff: The Second Report of a Seroprevalence Cohort Study from Iran. Amanat A, Bagheri Lankarani K, Honarvar B, Asmarian N, Shokripour M, et al. Shiraz E-Med J. 2022;23(9):e121681. doi: 10.5812/semj-121681.

Please put the ROC curve in the text WITH MORE ELABORATION ABOUT THE RESULTS ON THE DIAGRAM.

Author Response

All the suggestions were taken into consideration and the necessary changes were made in the text.

Comment: PLEASE ELABORATE SOME INTERNAL ORGAN DAMGE AND MORE INFORMATION  IN INTRODUCTION LINE 4.

THese ARTICLE CAN BE HELPFUL:

-Kidney diseases and COVID-19 infection: causes and effect, supportive therapeutics and nutritional perspectives. Hassan Askari, Nima Sanadgol, Asaad Azarnezhad, Amir Tajbakhsh, Hossein Rafiei, Ali Reza Safarpour, Seyed Mohammad Gheibihayat, Ehsan Raeis-Abdollahi, Amir Savardashtaki, Ali Ghanbariasad, Navid Omidifar.Heliyon,Volume 7, Issue 1,2021,e06008,ISSN 2405-8440, https://doi.org/10.1016/j.heliyon.2021.e06008.

Answer: Thanks for your observation. The comment was considered and included in the introduction. We have added the suggested reference

Comment: If we could have serologic data of patients to have a view of possible immunologic response on sign and symptoms, the article would have been stronger.

Such data for comparison can be found in these serial articles:

-Prevalence of anti-sars-cov-2 antibody in hospital staff in double-center setting: A preliminary report of a cohort study from Iran. Bagheri Lankarani K , Honarvar B, Omidifar N, Pakdin M, Moghadami M, et al. Shiraz E-Med J.22(3):e112681. doi: 10.5812/semj.112681.

-SARS-CoV-2 Antibodies in the Public Hospital Staff: The Second Report of a Seroprevalence Cohort Study from Iran. Amanat A, Bagheri Lankarani K, Honarvar B, Asmarian N, Shokripour M, et al. Shiraz E-Med J. 2022;23(9):e121681. doi: 10.5812/semj-121681.

Answer: Unfortunately, we do not have information on the immune response patients develop toward COVID. In the discussion section, when limitations are mentioned, a paragraph has been added about the results that could vary according to the genesis of antibodies in each patient, adding the suggested references.

Comment: Please put the ROC curve in the text WITH MORE ELABORATION ABOUT THE RESULTS ON THE DIAGRAM.

Answer: The ROC Curve diagram was added in the results section (figure 1)

Reviewer 3 Report

The current manuscript is an interesting report on characteristics of acute COVID 19 infection that can predict long term or chronic COVID symptoms. Although due to the pandemic the subject is well studied, however, the foresight in what charateristics can predict long term covid symptoms are novel and has potential to have potential clinical counseling or guidance for patients that fall in that category. 

Overall the manuscript is written and designed well,  some places the grammar can be improved.  Nevertheless does not hinder the significance of the paper. 

The conclusions draw in the manuscript are well explained in the discussion and are not a surprise from clinical standpoint. The Blood type having a protective effect against Covid was novel to be my understanding of this subject. 

Very minor comments are that in line 73 the reference has 2 different braces that can be rectified

Another comment was that reference 15 di do not have any mention of blood type having any relationship with covid, so that reference needs to be updated. 

Major feedback is that although the authors mention the limitation to be patient population representation and that they would like to extend this study to other locations as well to increase validity of their findings. Having some sort of treat recommendation or healthcare counseling recommendation etc for patients who have these strong charecteristic would add great significance to their finding. 

Author Response

All the suggestions were taken into consideration and the necessary changes were made in the text. 

Comment: The current manuscript is an interesting report on characteristics of acute COVID 19 infection that can predict long term or chronic COVID symptoms. Although due to the pandemic the subject is well studied, however, the foresight in what charateristics can predict long term covid symptoms are novel and has potential to have potential clinical counseling or guidance for patients that fall in that category.

Overall the manuscript is written and designed well,  some places the grammar can be improved.  Nevertheless does not hinder the significance of the paper

The conclusions draw in the manuscript are well explained in the discussion and are not a surprise from clinical standpoint. The Blood type having a protective effect against Covid was novel to be my understanding of this subject.

Very minor comments are that in line 73 the reference has 2 different braces that can be rectified

Another comment was that reference 15 di do not have any mention of blood type having any relationship with covid, so that reference needs to be updated.

Answer: Thanks for your comments. Fixed bugs mentioned about references. Reference 15 was replaced by another, according to the observation

Comment: Major feedback is that although the authors mention the limitation to be patient population representation and that they would like to extend this study to other locations as well to increase validity of their findings. Having some sort of treat recommendation or healthcare counseling recommendation etc for patients who have these strong charecteristic would add great significance to their finding.

Answer: Thank you very much for your comment. In the discussion section, possible measures are postulated that could help lower the incidence of long-term COVID, which are easily interpretable by clinicians.

Reviewer 4 Report

MANUSCRIPT REVIEW

JOURNAL:  MDPI     

TITLE:   Clinical characteristics in the acute phase of COVID-19 that predict long COVID: tachycardia, myalgias, severity, and use of antibiotic as main risk factors, while education and blood group B are protective

AUTHORS: Guzman-Esquivel et al 

ABSTRACT: No Comments

MANUSCRIPT:

Manuscript is well-written, and study is sufficiently detailed, and results are well explained. Few minor comments- 

1.     Strengths of this study can be explained and can be addressed briefly in a separate paragraph.  This is different from a survey study and patients were evaluated at clinic visit during the same time so less recall bias. The authors may summarize other strengths.  Similarly, only one limitation mentioned.  The symptom list did not include symptoms like sleeping difficulties, mental health symptoms – depression, anxiety etc. which have also been associated with long covid. 

2.     The lines in discussion – 296-298 “This agrees with the result in present study, where a higher number of symptoms in the acute phase, more hospitalizations and higher symptom severity score was associated with long covid” can be rephrased and added in conclusion which is currently very short. These findings are clinically relevant and can be a take home message. 

3.     Table 2 – “zing” to be corrected typing error

Author Response

All the suggestions were taken into consideration and the necessary changes were made in the text. 

Comment:

ABSTRACT: No Comments

Manuscript is well-written, and study is sufficiently detailed, and results are well explained. Few minor comments-

  1. Strengths of this study can be explained and can be addressed briefly in a separate paragraph. This is different from a survey study and patients were evaluated at clinic visit during the same time so less recall bias. The authors may summarize other strengths.  Similarly, only one limitation mentioned.  The symptom list did not include symptoms like sleeping difficulties, mental health symptoms – depression, anxiety etc. which have also been associated with long covid.

Answer: This limitation was incorporated into the discussions.

Comment: The lines in discussion – 296-298 “This agrees with the result in present study, where a higher number of symptoms in the acute phase, more hospitalizations and higher symptom severity score was associated with long covid” can be rephrased and added in conclusion which is currently very short. These findings are clinically relevant and can be a take home message.

Answer: Thanks for your observation. The information you mention now is also included in the conclusions.

Comment: Table 2 – “zing” to be corrected typing error

Answer: Thank you. The mistake was corrected